# Fulminant Myocarditis and Cardiogenic Shock Following COVID-19 Infection Versus COVID-19 Vaccination: A Systematic Literature Review

**DOI:** 10.3390/jcm12051849

**Published:** 2023-02-25

**Authors:** Maya E. Guglin, Aniekeme Etuk, Chirag Shah, Onyedika J. Ilonze

**Affiliations:** 1Division of Cardiovascular Medicine, Krannert Cardiovascular Research Center, Indiana University, Indianapolis, IN 46202, USA; 2Department of Medicine, Thomas Hospital-Infirmary Health, Fairhope, AL 36532, USA; 3Department of Medicine, Indiana University, Indianapolis, IN 46202, USA

**Keywords:** fulminant myocarditis, cardiogenic shock, COVID-19

## Abstract

Background: Myocarditis, diagnosed by symptoms and troponin elevation, has been well-described with COVID-19 infection, as well as shortly after COVID-19 vaccination. The literature has characterized the outcomes of myocarditis following COVID-19 infection and vaccination, but clinicopathologic, hemodynamic, and pathologic features following fulminant myocarditis have not been well-characterized. We aimed to compare clinical and pathological features of fulminant myocarditis requiring hemodynamic support with vasopressors/inotropes and mechanical circulatory support (MCS), in these two conditions. Methods: We analyzed the literature on fulminant myocarditis and cardiogenic shock associated with COVID-19 and COVID-19 vaccination and systematically reviewed all cases and case series where individual patient data were presented. We searched PubMed, EMBASE, and Google Scholar for “COVID”, “COVID-19”, and “coronavirus” in combination with “vaccine”, “fulminant myocarditis”, “acute heart failure”, and “cardiogenic shock”. The Student’s t-test was used for continuous variables and the χ2 statistic was used for categorical variables. For non-normal data distributions, the Wilcoxon Rank Sum Test was used for statistical comparisons. Results: We identified 73 cases and 27 cases of fulminant myocarditis associated with COVID-19 infection (COVID-19 FM) and COVID-19 vaccination (COVID-19 vaccine FM), respectively. Fever, shortness of breath, and chest pain were common presentations, but shortness of breath and pulmonary infiltrates were more often present in COVID-19 FM. Tachycardia, hypotension, leukocytosis, and lactic acidosis were seen in both cohorts, but patients with COVID-19 FM were more tachycardic and hypotensive. Histologically, lymphocytic myocarditis dominated both subsets, with some cases of eosinophilic myocarditis in both cohorts. Cellular necrosis was seen in 44.0% and 47.8% of COVID-19 FM and COVID-19 vaccine FM, respectively. Vasopressors and inotropes were used in 69.9% of COVID-19 FM and in 63.0% of the COVID-19 vaccine FM. Cardiac arrest was observed more in COVID-19 FM (*p* = 0.008). Venoarterial extracorporeal membrane oxygenation (VA-ECMO) support for cardiogenic shock was also used more commonly in the COVID-19 fulminant myocarditis group (*p* = 0.0293). Reported mortality was similar (27.7%) and 27.8%, respectively) but was likely worse for COVID-19 FM as the outcome was still unknown in 11% of cases. Conclusions: In the first series to retrospectively assess fulminant myocarditis associated with COVID-19 infection versus COVID-19 vaccination, we found that both conditions had a similarly high mortality rate, while COVID-19 FM had a more malignant course with more symptoms on presentation, more profound hemodynamic decompensation (higher heart rate, lower blood pressure), more cardiac arrests, and higher temporary MCS requirements including VA-ECMO. In terms of pathology, there was no difference in most biopsies/autopsies that demonstrated lymphocytic infiltrates and some eosinophilic or mixed infiltrates. There was no predominance of young males in COVID-19 vaccine FM cases, with male patients representing only 40.9% of the cohort.

## 1. Introduction

Myocarditis has been reported in association with COVID-19 infections and shortly after COVID-19 mRNA vaccination. We previously summarized all individual case reports in the English-language literature published in 2020 and found that myocardial involvement in COVID patients may present as a new and rapid decrease in left ventricular ejection fraction (LVEF), typically in the setting of bilateral lung damage. Moderate/large pericardial effusion was common, and cardiac tamponade occurred in 12.5% of patients. Cardiogenic shock developed in one third of the patients and mortality appeared to be high at 26% [1]. Later, we reported on acute myocarditis cases occurring within 60 days of mRNA COVID-19 vaccination that were reported in the English-language literature published in 2021 [2] and found that majority (87.1%) of patients were male, presenting typically with chest pain and elevated troponins. Cardiogenic shock was present in 11%, and mortality was 1.7% [2]. The outcomes of fulminant myocarditis with cardiogenic shock following COVID-19 infection and COVID-19 vaccination specifically in terms of inotrope use, mechanical circulatory support (MCS) use, and survival have not been well-characterized.

The pathophysiology of myocardial injury in both COVID-19 infection and post COVID-19 vaccination is not well understood. In the case of COVID-19, it is thought that both direct viral damage and hyperimmune response may play a role [3]. Because COVID-19 virus was rarely found in the cardiomyocytes [4], an exaggerated immune response is a plausible mechanism. In the case of mRNA vaccine, which does not contain the virus, immune response also may play a role. It was previously suggested that COVID-induced and COVID vaccine-induced myocarditis may share a common pathology [5].

We therefore wanted to compare clinical and pathological features of myocarditis in both entities. To avoid diagnostic ambiguity, we focused only on cases of fulminant myocarditis with a significant drop in left ventricular ejection fraction (LVEF) and hemodynamic compromise requiring pharmacological support (vasopressors, inotropes), or MCS, or both.

## 2. Methods

### 2.1. Study Design

This study was a systematic review of the literature conducted following PRISMA guidelines [6] to retrieve publications containing data regarding clinical presentations, course, and outcomes of fulminant myocarditis +/− cardiogenic shock following COVID-19 infection and COVID-19 vaccination. Registration of a review protocol was unnecessary because data contained in the published literature were used to conduct this study.

### 2.2. Eligibility Criteria

The publications included were full-length manuscripts retrieved with our search that contained data on one or more patients, who were 18 years old or older, with either a positive test for COVID-19 or COVID-19 vaccination with fulminant myocarditis diagnosed by all of the following characteristics: (1) new systolic dysfunction reported as a decrease in left ventricular ejection fraction (LVEF); (2) hemodynamic instability requiring pharmacological support (vasopressors, inotropes) or mechanical circulatory support (MCS), or both. Publications were excluded if these were written in languages other than English, contained pediatric population data (patients younger than 18 years old), or had insufficient data on individual patients.

### 2.3. Search Method

Following PRISMA guidelines [6], a systematic search of the literature was conducted using PubMed, Google Scholar, and EMBASE. The keywords used were “COVID”, “COVID-19”, and “coronavirus” in combination with “vaccine”, “fulminant myocarditis”, “acute heart failure”, and “cardiogenic shock”. In relevant papers, the list of references was manually searched as well. The search was limited to the articles published in English from 1 December 2020–31 August 2022. All the identified publications were screened to exclude duplicates by comparing titles, authors, and digital object identifiers. After removing duplicates, all the remaining publications were screened for the exclusion criteria by reading titles and abstracts. After removing publications that met the exclusion criteria, the remaining publications were further screened for the inclusion and exclusion criteria by reading the full-text publications.

The list of references for each relevant publication was manually examined. Through this searched method, the publications to be included in the analysis were identified (Figure 1). To assess for reproducibility, the search was conducted independently by two members of the working group, and results were manually compared.

### 2.4. Data Extraction Process

The included publications were analyzed in a qualitative manner for authors’ names, date of publication, medical center where they were treated, country, and the timeline of the events. These publications were used to identify our subjects of interest. Once identified, subjects were labeled and their data extracted. These data were used to perform quantitative analyses of demographics, comorbidities, biomarkers, symptoms on presentation, imaging studies, type of vaccine (where applicable), changes of LVEF, wall motion abnormalities, presence of cardiogenic shock, use of temporary mechanical circulatory support devices (MCS), and outcomes (survival to discharge). The time of the onset of symptoms was limited to 60 days from either COVID-19 positive test or COVID-19 vaccine.

### 2.5. Statistical Analysis

Data extracted from the publications were grouped into continuous or categorical variables for analysis. Prior to choosing the appropriate test, the distribution of the data was analyzed, particularly for continuous variables. For continuous variables, the Student’s t-test was used when appropriate, with attention given to equality of variances. However, the data were not normally distributed for most continuous variables, so the Wilcoxon Rank Sum Test was used for statistical comparisons. For categorical variables, the χ^2^ statistic was used to make statistical comparisons. When sample size for entries within specific subgroups was less than 10, the Fisher’s exact test was utilized. Statistical significance was determined at alpha levels of 0.05 and 0.01. All statistical analyses were conducted using SAS version 9.4 (SAS Institute, Cary, NC, USA).

## 3. Results

### 3.1. Literature Search

The search identified 634 (482 + 152) publications. After removing duplicates and screening for exclusion and inclusion criteria, 100 publications were included in the analysis (Figure 1).

### 3.2. Patient Characteristics

#### 3.2.1. Fulminant Myocarditis in COVID-19 Positive Patients

There were 73 cases of fulminant myocarditis in COVID-19 positive patients, comprising 40 (54.8%) males and a mean age of 45.1 ± 15.8 years (Table 1). On presentation, fever (*n* = 37, 50.7%), shortness of breath (*n* = 35, 47.9%), and chest pain (*n* = 10, 13.7%) were the commonest symptoms. On physical examination, tachycardia and hypotension were common with a mean heart rate of 125.1 ± 21.9 beats per minute, a mean systolic blood pressure of 86.7 ± 18.8 mmHg, and a mean diastolic blood pressure of 54.6 ± 14.9 mmHg. Leukocytosis and lactic acidosis were common with a mean white blood count of 19.200 ± 980/mm^3^ and serum lactate level of 7.3 ± 4.2 mmol/L. Diffuse bilateral pulmonary infiltrates, characteristic of COVID-19 infection, were present in 31 (42.5%) patients.

The lowest recorded LVEF was 19.2 ± 8.9%, and it recovered to 56.5 ± 8.0%. Pericardial effusion on echocardiography was reported in 26 (35.6%) patients, including tamponade in 9 (12.3%) cases. In 25 (34.3%) cases, coronary angiogram was performed and was negative for obstructive lesions in all cases. Cardiac magnetic resonance imaging was done in 13 (17.8%) patients and was consistent with myocarditis. In terms of management, 68.5% received steroids, 26.0% received antiviral agents, and 37.0% received antibiotics.

Vasopressors and inotropes were used in 51 (69.9%) patients and MCS in 55 (75.3%), including intra-aortic balloon pump (12/16.4%), Impella^®^ devices (Abiomed, Danvers, MA, USA) (5/6.8%), and venoarterial extracorporeal membrane oxygenation (VA ECMO) in (38) 52.1% of cases. In the course of the disease, cardiac arrest occurred in 18 (24.7%) patients. Out of this cohort, 44 (60.3%) survived to discharge, 21 (28.8%) died, and in 8 (11.0%), the outcome was still unknown at the time of publication.

Pathology findings were available in 25 (34.2%) patients who had either undergone biopsy or autopsy. In 17 patients, the infiltrates in the myocardium were predominantly lymphocytic; in four patients, the infiltrates were eosinophilic; and in two patients, the infiltrates were dominated by macrophages. In one patient, the biopsy was negative. Cellular necrosis was present in 11 patients (44.0%). In four cases, the COVID-19 viral particles were identified in either the interstitial cells or in the cardiomyocytes themselves, while in seven patients, the presence of COVID RNA was confirmed by genomic studies. In four cases, there were microthrombi reported in the myocardium.

#### 3.2.2. Fulminant Myocarditis after COVID-19 Vaccination

There were 27 cases of fulminant myocarditis in patients within 60 days of COVID-19 vaccination, comprising 11 (40.7%) males and a mean age of 50.3 ± 16.4 years (Table 2). The Pfizer vaccine was used in 15 (55.5%) cases, of which the myocarditis developed after the first dose in eight (53.3% of vaccinated with the Pfizer product), after the second dose in seven (25.9%), after the booster in one, and were unreported in the rest of the patients. The Moderna vaccine was used in six (22.2%) patients, and myocarditis developed in two patients after the first dose, three patients after the second dose, and one after the booster. Vero cell, AstraZeneca, and Janssen vaccines were used in two patients each. Although we limited the timing from the last dose of the vaccine to the onset of symptoms to 60 days, the longest interval between the vaccine and the symptoms in our cohort was 24 days, with a mean of 6.8 ± 5.4 days.

Fever (*n* = 14, 51.9%), chest pain (*n* = 11, 40.7%), and shortness of breath (*n* = 10, 37.0%) were the most common symptoms. On physical examination, tachycardia with a mean heart rate of 111.3 ± 24.7 beats per minute and hypotension with a systolic blood pressure of 90.1 ± 17.0 mmHg and a diastolic blood pressure of 62.9 ± 12.7 mmHg were common. Leukocytosis and lactic acidosis were common with a mean white blood count of 14.300 ± 1020/mm^3^, and serum lactate level was 9.6 ± 8.4 mmol/L. Pulmonary edema or signs of congestion on chest X-ray were present in five (18.5%) patients.

The lowest recorded LVEF was 20.8 ± 9.7%, and it recovered to 56.1 ± 7.1%. Pericardial effusion on echocardiography was reported in 12 (44.4%) patients. In 21 (77.8%) patients, coronary angiogram was performed, and it was negative for obstructive lesions in all cases. Cardiac magnetic resonance imaging was done in nine cases (33.3%) and was consistent with myocarditis in eight of them. In one case, wherein cardiac MRI was negative for myocarditis, characteristic changes of myocarditis were seen on biopsy [69].

In terms of management, 66.7% received steroids. Vasopressors and inotropes were used in 17 (63.0%) cases and MCS in 29 (>100%), because more than one device was used in seven cases, including intra-aortic balloon pump (8/30.0%), Impella^®^ (Abiomed, Danvers, MA, USA) (9/33.3%), and venoarterial extracorporeal membrane oxygenation (VA ECMO) in (12) 44.4% of cases. In the course of the disease, cardiac arrest occurred in seven (25.9%) patients. Out of this cohort, 19 (70.3%) survived to discharge, and 21 (28.7%) died.

Pathological diagnosis was available in 23 (85.2%) cases. Infiltrates were lymphocytic in 14 cases, mixed lymphocytic and eosinophilic in 3 cases, predominantly eosinophilic in 3 cases, and giant cells were present in one case. In one case, infiltrates consisted of neutrophils and histiocytes, and in one case, biopsy was negative. Cellular necrosis was present in 11/23 (47.8%) cases. There were no reports of microthrombotic injuries in this cohort.

#### 3.2.3. Comparison of COVID-19 FM with COVID-vaccine FM

In comparison of COVID-19 and COVID vaccine related FM, patients with COVID myocarditis had a higher heart rate (125.1 ± 21.9 beats per minute vs. 106.8 ± 30.7 beats per minute, *p* = 0.021), lower diastolic and mean blood pressures (54.6 ± 14.8 mmHg vs. 62.9 ± 12.7 mmHg, *p* = 0.031; and 62.1 ± 17.0 mmHg vs. 69.7 ± 17.mmHg, *p* = 0.049, respectively), a trend towards higher brain natriuretic peptide concentration (11,468.3 ± 18,237.7 pg/dL vs. 947.0 ± 6144.8 pg/dL, *p* = 0.056), and a more frequent occurrence of pulmonary infiltrates on chest X-ray, while patients with recent vaccination presented more often with generalized symptoms of malaise and fatigue (Table 3). COVID-19 FM patients more often experienced cardiac arrest and used VA ECMO support, Nevertheless, there was no mortality difference (Table 3).

## 4. Discussion

In this systematic review, we summarized the features, including demographics, symptoms, clinical characteristics, treatment, and outcomes of fulminant myocarditis occurring in the setting of COVID infection versus recent vaccination against COVID. While there were several papers trying to compare COVID versus COVID-vaccine related myocarditis, this is the first analysis, to our knowledge, focusing on the most severe forms of myocarditis with low LVEF and the need for vasopressors with or without MCS, suggestive of severe fulminant myocarditis with cardiogenic shock.

We found that:Patients with COVID-19 FM, in comparison with COVID-19 vaccine FM, had higher a heart rate, lower diastolic and mean blood pressures, and a higher prevalence of pulmonary infiltrates on chest X-ray, and they presented more commonly with fever and shortness of breath and less commonly with generalized symptoms.Despite differences in presentation, mortality in COVID-19 FM myocarditis and COVID-19 vaccine FM was similar (27.7% and 27.8%, respectively).There was no difference in pathology between COVID-19 FM myocarditis and COVID-19 vaccine FM, with most biopsies/autopsies demonstrating lymphocytic infiltrates and some eosinophilic or mixed infiltrates.There was no predominance of young males in COVID-19 vaccine FM cases. Females represented 59.3% of this cohort.

Patients with COVID-19 FM more frequently presented with fever and shortness of breath. Heart rate was also higher, while diastolic and mean blood pressures were lower, in COVID-19 FM. Patients in the COVID cohort experienced cardiac arrest and required ECMO support more commonly than patients with recent vaccination. They also had more pulmonary infiltrates on chest radiogram. The difference in presentation therefore can be explained by concomitant lung disease in the setting of acute respiratory infection.

Multiple papers describing vaccine-induced myocarditis noticed that affected individuals were predominantly young males in their teens or early 20s [2,89]. Surprisingly, we did not see that in the case of fulminant myocarditis as 59.3% of our cohort were females. In published reports on COVID-19 vaccine induced myocarditis, males constituted over 65% [90]; male to female ratio was reported as 3:2 [91] or even 14:1 [92]. According to another report, male sex increased the odds of myocarditis 4.4 times [93]. In our review, only 40.1% of patients were male, which also suggests a principally different mechanism in FM compared to that in usual clinically mild myocarditis.

This observation may mean that patients’ predisposition and not the virus or the vaccine determines catastrophic events with rapid and profound deterioration of the contractile function of the myocardium. This may explain similar outcomes. Differences in clinical presentation may be due to concomitant respiratory infection in COVID patients. Besides, the true mortality rate in the COVID-19 FM remains unknown. Because of the urge to publish any new evidence at the height of the pandemic, the outcomes are unknown in 11% of cases we collected. They were published while the patients still remained hospitalized. At the same time, in the COVID-19 vaccine FM, outcomes were reported in all cases. This allows the speculation that true mortality in COVID-19 FM may be higher.

Histologically, the data showed predominantly lymphocytic myocarditis in the majority of cases of fulminant myocarditis and cardiogenic shock of both etiologies. There was evidence of COVID-19 viral particle or viral genome seen on myocardial samples in some patients with COVID-19 FM. We did not see any cases with the evidence of severe acute respiratory syndrome coronavirus 2 spike protein in COVID-19 vaccine FM although it was reported after the vaccination before [75]. Microthrombi in the setting of COVID-19 myocarditis were quite rare but were seen in four cases. Further, some very unusual platelet thrombi have been reported [19].

The differences between COVID-induced and COVID-vaccine induced myocarditis may be explained by: (1) different pathophysiology or (2) same pathophysiology; however, patients were more symptomatic and hemodynamically unstable because they may have other manifestations of acute respiratory infection (they present with fever and shortness of breath more commonly than in the case of a vaccine). The mechanism of myocardial damage in either case is not well established. Direct viral infection of cardiomyocytes is one potential mechanism [94]. The COVID-19 virus is a large, enveloped, single-stranded RNA virus that binds to the angiotensin-converting enzyme (ACE)2 receptor and enters the cell. ACE2 receptor is present in multiple organs, but the expression in different tissues is different, and the cardiac muscle is rich with ACE2 receptors [95]. Nevertheless, in a few cases, the viral particles were found in the myocardium [57,96], but in other cases they were not. Moreover, in the case we reported [64], parvovirus was found in the cardiac muscle. Potentially, COVID-19 may make patients more susceptible to myocarditis caused by other pathogens. Shah et al. noticed a high (22%) co-infection rate in patients with coronavirus [54,97].

The other possibility involves the virus triggering immune response in the form of cytokine storm releasing IL-6, IL-10, and TNF-alpha, causing myocarditis. The intensity of the storm is linked to the severity of infection [98] or represent an exaggerated hyperimmune response of the host. Further, there is no unity behind the understanding of pathophysiology of vaccine-induced myocarditis. Since the vaccines do not contain the virus, the mechanisms may include hypersensitivity reaction and genetic susceptibility (variants in genes encoding human leukocyte antigen, desmosomal, cytoskeletal, or sarcomeric proteins, making an individual more susceptible to myocardial damage). Spike protein is present in mRNA vaccines. The viral spike protein, produced by the cell after mRNA- vaccine entry, induces an adaptive immune response to identify and destroy viruses that express the spike protein. Vaccine-induced spike-protein IgG antibodies thereby neutralize the virus. The antibodies to SARS-CoV-2 spike glycoproteins may cross-react with myocardial contractile proteins, including myocardial α- myosin heavy chain [99]. Spike S protein was detectable 2 weeks after the first dose of the BNT162b2 (Pfizer BioNTech) vaccine, with a significant increase 2 weeks after the second dose. Further, antibodies to the SARS-CoV-2 spike S protein were increased on day 14 after the second dose [100]. Moreover, the Pfizer vaccine stimulates the spike S protein specific T cell responses, which were detectable up to 4 weeks after the second dose. Therefore, the vaccine can produce both cellular and humoral immune response. The amplified immune response to the spike protein seems to be the most likely pathophysiology behind both COVID-induced and COVID vaccine-induced fulminant myocarditis.

It remains unclear why most individuals developing post-vaccine myocarditis experienced self-limited symptoms of chest pain and demonstrated mild troponin elevation, while others presented with overwhelming hemodynamic compromise or cardiac arrest. The similarity of manifestations between COVID-19 FM and COVID-19 vaccine FM suggests that such hyperreaction to either infection or the vaccine characterizes the individual reactivity more than the trigger. The devastating nature of fulminant myocarditis of either origin justifies future clinical, immunological, and genetic study in order to understand, treat, and ultimately prevent such response in the future.

Pathology was available only in some reports on COVID-19 FM, but in almost all cases of COVID-19 vaccine FM.

## 5. Conclusions

In the first series to retrospectively assess fulminant myocarditis associated with COVID-19 infection versus COVID-19 vaccination, we found that COVID-19 FM has a more malignant course with more symptoms on presentation, more profound hemodynamic decompensation (higher heart rate and lower blood pressure), more cardiac arrests, and more temporary MCS requirements including VA-ECMO. Nevertheless, mortality was similarly high at 27.7% and 27.8%, respectively. In terms of pathology, there was no difference in most biopsies/autopsies that demonstrated lymphocytic infiltrates and some eosinophilic or mixed infiltrates.

There was no predominance of young males in COVID-19 vaccine FM cases. Males represented only 40.1% of this cohort.

## 6. Limitations

The retrospective and descriptive nature of our study was a limitation. Since we limited our literature search from 1 December 2021–31 August 2022, several cases published in 2020 and later in 2022 were not included in the analysis. Definition of myocarditis was also based on local clinical expertise and diagnostic modalities. The criteria for cardiogenic were limited to any inotrope/vasopressor use or use of MCS, and we may have overestimated the incidence of cardiogenic shock. Due to inadequate data in some patients, some may have had concomitant pericarditis (myopericarditis), which may have a variable presentation from frank myocarditis. Some patients included in this series may have had an alternative diagnosis, although the severe fulminant myocarditis suggests otherwise. Pathology was available only in some reports on COVID-19 FM, but in almost all cases of COVID-19 vaccine FM.

## Figures and Tables

**Figure 1 jcm-12-01849-f001:**
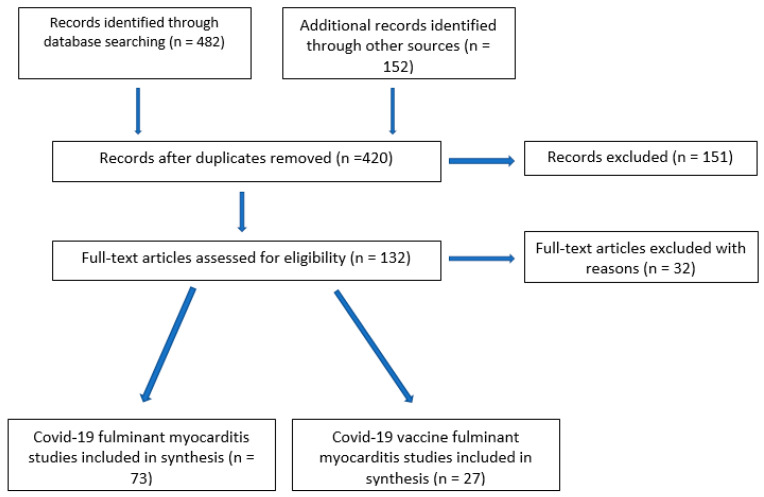
The literature search: the flow diagram.

**Table 1 jcm-12-01849-t001:** Cases of COVID-19 fulminant myocarditis.

Authors	Country	Age, Years	Sex (M = 1; F = 2)	Vasoactive Agents (Yes = 1; No or not Reported = 2)	Intra-Aortic Balloon Pump (Yes = 1; No = 2)	Impella (Yes = 1; No = 2)	ECMO (Yes = 1; No = 2)	Survival to Discharge (Yes = 1; No = 2)	Treatment	Biopsy/Infiltrate
Afriyie [7]	USA	27	1	1				2	Remdesivir, Steroids	
Albert [4]	USA	49	1	2			1	1	Tocilizumab, Steroids, IVIG	Lymphocytes
Aldeghaither [8]	Canada	39	2	1			1	1	Steroids	Eosinophils
Aldeghaither [8]	Canada	25	1	1			2	1	Steroids, IVIG, Anakinra (IL-antagonist)	Eosinophils
Aldeghaither [8]	Canada	21	1	1			1	1	Steroids, Anakinra, IVIG	Lymphocytes
Almanza [9]	Colombia	75	2	1				2	Tocilizumab	Lymphocytes
Anupama [10]	USA	66	1	1				1	Hydroxy-chloroquine, IV diuretics, IVIG, convalescent plasma	
Bergman [11]	USA	53	1	1			1	1	Hydroxy-chloroquine, Sarilumab (IL-6 inhibitor)	
Bernal-Torres [12]	Colombia	38	2	1				1	Steroids, IVIG, Hydroxy-chloroquine, Lopinavir/Ritonavir	
Bhardwaj [13]	USA	22	1	2			1	1	Steroids, Remdesivir	
Bhardwaj [13]	USA	53	2	2			1	1	Steroids, Remdesivir, Convalescent Plasma	
Bhardwaj [13]	USA	28	2	2			1	2	Steroids, Remdesivir, Tocilizumab, IVIG	
Bhardwaj [13]	USA	27	2	2			1	1	Steroids, Remdesivir, IVIG	
Bhardwaj [13]	USA	46	1	2			1	1	Steroids, Remdesivir, IVIG	
Bhardwaj [13]	USA	68	1	2			1	2	Steroids	
Bhardwaj [13]	USA	26	2	2			1	1	Steroids	
Bhardwaj [13]	USA	66	1	2			1	1	Steroids	
Bhardwaj [13]	USA	24	1	2			1	1	Steroids	
Bulbul [14]	Qatar	49	2	1			1	1	Hydroxy-chloroquine, Oseltamivir, Teicoplanin, Steroids, Tocilizumab, Lopinavir, Ritonavir	
Chitturi [15]	USA	65	2	1			2	1	Remdesivir, Steroids, Tocilizumab	
Coll [16]	USA	39	1	1	2	2	2	1	Steroids, IVIG	
Coyle [17]	USA	57	1	1				1	Hydroxy-chloroquine, Steroids, Colchicine, Tocilizumab	
Craver [18]	USA	18	1	2				2		Eosinophils
Fiore [19]	Italy	45	1	1	1	2	2	1	Hydroxy-chloroquine, Anakinra,	Lymphocytes
Fried [20]	USA	38	1	1			1		Hydroxy-chloroquine	
Fried [20]	USA	64	2	1	2					
Garau [21]	Belgium	18	2	1	1		1	1	Steroids, Hydroxy-chloroquine, IVIG	Lymphocytes
Garot [22]	France	18	1	1				1	Hydroxy-chloroquine	
Gauchotte [23]	France	69	1	1			1	2		Lymphocytes
Gaudriot [24]	France	38	1	1	1	2	1	1	Steroids, Mycophenolate Mofetil, Cyclosporine	Lymphocytes
Gay [25]	USA	56	1	1		2	1	1	Steroids, Tocilizumab	
Ghafoor [26]	USA	54	2	1	2	2	1	2		
Gill [27]	USA	65	2	1	1		2	2		
Gill [27]	USA	34	2	2			1	1	Colchicine, Steroids	
Gomez [28]	Spain	53	1	1				2		
Hamdan [29]	UAE	45	2	1			1	1	Steroids	
Hu [30]	China	37	1	1				1	Steroids, IVIG.	
Hussain [31]	USA	51	1	1					Hydroxy-chloroquine, Azithromycin, Colchicine, Steroids, Remdesivir	
Irabien-Ortiz [32]	Spain	59	2	1	1		1		Lopinavir/Ritonavir, IVIG, Steroids	
Ishikura [33]	Japan	35	1	1	1		1	1	IVIG, Steroids	Lymphocytes
Jacobs [34]	Belgium	48	1	1			1	2	Hydroxy-chloroquine	Lymphocytes
Kallel [35]	Morocco	26	1	1	2	2	2	1		
Khatri [36]	USA	50	1	1			2	2	Hydroxy-chloroquine, IVIG, Steroids	
Li [37]	USA	60	1	1				1	Hydroxy-chloroquine, Steroids, IVIG	
Mansoor [38]	USA	72	2	1				2	Chloroquine, IVIG, Steroids	
Menter [39]	Switzer-land	47	2	1	2	2	2	2		Lymphocytes, Neutrophils
Milla-Godoy [40]	USA	45	2	1				2	Hydroxy-chloroquine	
Nakatani [41]	Japan	49	1	1				2	IVIG, Steroids	Lymphocytes
Naneishvili [42]	UK	44	2	1				1	Steroids	
Nwaejike [43]	UK	39	1	1			1	1		
Okor [44]	USA	72	2	1	2	2	2	2	Steroids	
Othenin-Girard [45]	Switzer-land	22	1	2			1	1	Steroids, Tocilizumab, IVIG, Cyclo-phosphamide, Rituximab	Lymphocytes
Papageorgiou [46]	Sweden	43	1	1			1	1	Steroids, Colchicine	None
Purdy [47]	USA	53	1	1				1	Steroids, Hydroxy-chloroquine	
Rajpal [48]	USA	60	2	1			1	1	Steroids	Lymphocytes
Richard [49]	USA	28	2	1		1		1	Steroids	
Ruiz [50]	USA	35	2	2		1		1	IVIG, Remdesivir, Steroids	
Salamanca [51]	Spain	44	1	1	1		1		Steroids, Tocilizumab, Hydroxy-chloroquine, Lopinavir/Ritonavir	Lymphocytes
Sampaio [52]	Brazil	45	2	1			1	1	Tocilizumab, IVIG, Convalescent Plasma, Steroids	
Sanchez [53]	Spain	42	2	2	1		1	2		
Shah [54]	Qatar	19	1	1					Steroids, Hydroxy-chloroquine, Tocilizumab, IVIG	
Shahrami [55]	Iran	78	1	1				2	Steroids, IVIG, Hydroxy-chloroquine	
Shen [56]	USA	43	1	1	1	2	2	1	IVIG	
Tavazzi [57]	Italy	69	1	1	1		1	2	Lopinavir/Ritonavir, Hydroxy-chloroquine, Steroids	Lymphocytes
Thaker [58]	USA	42	2	2	1	2	2	1	Tocilizumab	
Thaker [58]	USA	42	2	2			1		Steroids, Tocilizumab, IVIG	
Thomson [59]	Australia	39	2	1			1	2	Steroids, IVIG	Macrophages
Trpkov [60]	Canada	62	2	2				1	Steroids, Anakinra	
Verma [61]	USA	48	2	2		1	1	1	Steroids, IVIG, Remdesivir, Tocilizumab	Macrophages
Veronese [62]	Italy	51	2	2	1		1	1	Steroids	Lymphocytes
Yalcinkaya [63]	Turkey	29	1	1						Eosinophils
Yeleti [64]	USA	25	2	2	2		1	1	Steroids Remdesivir, Convalescent Plasma	Lymphocytes
Zeng [65]	China	63	1	2				2	Antiviral	

**Table 2 jcm-12-01849-t002:** Cases of COVID-19 vaccine fulminant myocarditis.

Authors	Country	Age, Years	Sex (M = 1; F = 2)	Type of Vaccine (Pfizer =1; Moderna = 2; Vero Cell =3; AstraZeneca = 4; Janssen = 5)	Dose to Symptoms (Days)	Number of Doses	Vasoactive Agents (Yes = 1; No or not Reported = 2)	Intra-Aortic Balloon Pump (Yes = 1; No = 2)	Impella (Yes = 1; No = 2)	ECMO (Yes = 1; No = 2)	Survival to Discharge (Yes = 1; No = 2)	Treatment	Biopsy/ Infiltrate
Abbate [66]	USA	27	1	1	2	2	1	2	2	1	2	Steroids	
Abbate [66]	USA	34	2	1	9	1	1	2	2	1	1	Steroids	Lymphocytes
Agdamag [67]	USA	80	2	1	7	1	1	2	1	2	1	Steroids	Eosinophils
Ameratunga [68]	New Zealand	57	2	1	1	1	2				2		Eosinophils
Araki [69]	Japan	53	2	1	4	2	1	1	1	1	1	Steroids	None
Brage [70]	Spain	62	2	2	1	3	1	2	1	2	1	Steroids	
Choi [71]	Korea	22	1	1	5	1	2	2	2	2	2		Neutrophils, Histiocytes
Cui [72]	China	57	2	3	4	1		1	2	2	1	Steroids	Lymphocytes
Cui [72]	China	63	1	3	4	1	2	1	2	2	1	Steroids	Lymphocytes
Hoshino [73]	Japan	27	1	2	8	1	1		1	1	2	Steroids, IVIG	Lymphocytes
Kadwalwala [74]	USA	38	1	2	2	1	1	2	1	2	1	Steroids	Lymphocytes
Kawano [75]	Japan	60	2	1	24	2	1	2	1	1	1	Steroids	Lymphocytes
Kazama [76]	Japan	48	2	2	7	2	1	1	1	1	1		Lymphocytes
Kim [77]	Korea	63	2	4	1	1	2	2	2	1	2		Lymphocytes
Kimura [78]	Japan	69	1	1	7	1	1	1	1	1	1	Steroids	Lymphocytes, Eosinophils
Koiwaya [79]	Japan	77	1	1	8	1	2	1	2	1	1		Lymphocytes
Lim [80]	South Korea	38	2	1	7			2	2	1	1		Lymphocytes
Nassar [81]	USA	70	2	5	2		1	2	2	2	2		
Oka [82]	Japan	50	1	1	10	2	1	2	2	2	1	Steroids	Lymphocytes
Olmos [83]	Canada	49	2	1	6	2	1			1	1	Steroids	Lymphocytes, Eosinophils
Sung [84]	USA	63	1	1	7	2	1	1			1	Steroids	Giant cells
Ujieta [85]	USA	62	2	5	4		1				2	Steroids	Lymphocytes
Verma [86]	USA	45	2	1	10	1	1	2	2	2	1		Lymphocytes, Eosinophils
Verma [86]	USA	42	1	2	14	2	2				2		Lymphocytes, Eosinophils
Wu [87]	Taiwan	44	2	4	2		1	2	2	2	1		
Yamamoto [88]	Japan	41	1	2	19	2	2		1	1	1	Steroids	Lymphocytes
Yamamoto [88]	Japan	18	2	1	9	1	2	1	2	2	1	Steroids	Lymphocytes

**Table 3 jcm-12-01849-t003:** Comparison of clinical features and outcomes of fulminant myocarditis and cardiogenic shock following COVID-19 and COVID vaccination.

	COVID-19 Fulminant Myocarditis			COVID-19 Vaccine Fulminant Myocarditis			*p*-Value
Continuous Variables							
	*n*	Mean	SD	*n*	Mean	SD	*p*-Value
Heart rate, beats per minute	29	125.1	21.9	18	106.8	30.7	0.0211
Diastolic blood pressure, mmHg	28	54.6	14.8	17	62.9	12.7	0.0312
Mean arterial pressure, mmHg	33	62.1	17.0	18	69.7	17.6	0.0497
BNP, pg/dL	16	11,468.3	18,237.7	9	947.0	614.8	0.0558
Age	73	45.1	15.8	27	50.3	16.4	0.1490
Systolic blood pressure, mmHg	30	86.7	18.8	18	90.2	17.0	0.5287
White blood cells, peak	33	19.2	9.8	12	14.3	10.2	0.1457
Lactate	33	7.3	4.2	8	9.6	8.4	0.7174
C-reactive protein	55	94.2	350.8	14	33.6	68.1	0.0552
Peak troponin I	20	5.7 × 10^8^	2.5 × 10^9^	13	6.4 × 10^8^	2.3 × 10^9^	0.5311
LVEF at admission	63	19.3	8.9	24	20.8	9.7	0.5513
Length of stay total	41	26.9	20.0	15	27.1	25.4	0.5661
Categorical variables							
	*n*	Yes	No	*n*	Yes	No	
Fever	48	38	10	26	14	12	0.0229
Shortness of breath	42	35	7	24	10	14	0.0005
Pulmonary infiltrates on chest X-ray	73	34	39	27	5	22	0.0117
General symptoms	73	10	63	27	13	14	0.0008
Cardiac arrest	22	18	4	23	7	16	0.0008
VA-ECMO	49	38	11	23	12	11	0.0293
Gender	73	40 (M)	33 (F)	27	11 (M)	16 (F)	0.2120
Intubation	73	30	43	27	9	18	0.4798
Right ventricular dysfunction	73	26	47	27	3	24	0.0165
Chest pain	73	10	63	27	11	16	0.0032
Pericardial effusion	73	15	58	27	12	15	0.0169
Intra-aortic balloon pump	73	12	61	27	8	19	0.1432
Survival	73	44	29	27	19	8	0.3532
Gastrointestinal symptoms	73	16	57	27	4	23	0.4305

BNP—brain natriuretic peptide, VA-ECMO—venoarterial extracorporeal membrane oxygenation, SD—standard deviation.

## Data Availability

All studies are published and publicly available.

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
