# Peer review of "Fulminant Myocarditis and Cardiogenic Shock Following COVID-19 Infection Versus COVID-19 Vaccination: A Systematic Literature Review"

_jcm, 2023, doi:10.3390/jcm12051849_

Round 1
Reviewer 1 Report
The authors compared the two fulminant myocarditis groups caused by different etiology. The focus point is interesting, but there are some major and minor problems that need to be resolved.
Major:
1) This study’s main aim is to compare clinical and pathological features between the two groups. Therefore, I recommend the authors show the comparison of each indicator and data one by one in the result section, not the discussion section. This style would make the abstract more attractive.
2) The main table (Table 3) should include all information and data you showed in the result section. The authors should add some data, for example, age sex, medical treatment (steroid), pathological findings, etc.
3) The major issues the authors should explain in the discussion section are (a) why the clinical symptom is significantly severe in the CPVID-19 infection-induced fulminant myocarditis, and (b) why the mortality rate is comparable in the two groups. The authors are discussing (a), but not (b) deeply.
4) Figure 1 does not look like a flow chart commonly used in the clinical study. We cannot follow the accurate sample number included or excluded by the criteria. The authors need to check clinical articles that have a flow chart of patient enrollment as a reference (for example, PMID: 30626332).
5) Is there any special reason that the authors did not include the information on the medical treatment in Table 2? The authors need to keep consistency with Table 1.
6) What is the clinical implication of this study? The authors need to emphasize it.
Minor:
1) The conclusion of the abstract should be clear and concise. Detailed data and values should not be included in the section.
2) There is no information on the EMBASE in the abstract.
3) I cannot understand the sentence of the abstract ‘The Student’s t-test was used for continuous variables and the χ2 statistic was used for categorical variables while the Wilcoxon Rank Sum Test was used for statistical comparisons’. T-test and χ2 statistics are also analysis for the comparison.
4) There are some extra spaces between words and sentences.
Author Response
Reviewer 1
Comments and Suggestions for Authors
The authors compared the two fulminant myocarditis groups caused by different etiology. The focus point is interesting, but there are some major and minor problems that need to be resolved.
Major:
1) This study’s main aim is to compare clinical and pathological features between the two groups. Therefore, I recommend the authors show the comparison of each indicator and data one by one in the result section, not the discussion section. This style would make the abstract more attractive.
Response:
We added the following section to the Results:
Comparison of COVID-19 FM with COVID-vaccine FM
In comparison of COVID-19 and COVID vaccine related FM, patients with COVID myocarditis had higher heart rate (125.1±21.9 beats per minute vs 106.8±30.7 beats per minute, p=0.021), lower diastolic and mean blood pressure (54.6±14.8mmHg vs 62.9±12.7mmHg, p=0.031, and 62.1±17.0mmHg vs 69.7±17.mmHg, p=0.049, respectively), a trend towards higher brain natriuretic peptide concentration (11,468.3±18,237.7pg/dL vs 947.0±6144.8pg/dL, p=0.056), andmore frequent pulmonary infiltrates on chest X-ray, while patients with recent vaccination presented more often with generalized symptoms of malaise and fatigue (Table 3). COVID-19 FM patients more often experienced cardiac arrest and used VA ECMO support, Nevertheless, there was no mortality difference (Table 3).
2) The main table (Table 3) should include all information and data you showed in the result section. The authors should add some data, for example, age sex, medical treatment (steroid), pathological findings, etc.
Response:
We added variables to Table 3.
3) The major issues the authors should explain in the discussion section are (a) why the clinical symptom is significantly severe in the CPVID-19 infection-induced fulminant myocarditis, and (b) why the mortality rate is comparable in the two groups. The authors are discussing (a), but not (b) deeply.
Response:
We added the following consideration:
Besides, the true mortality rate in the COVID-19 FM remains unknown. Because of the urge to publish any new evidence at the height of the pandemic, the outcomes are unknown in 11% of cases we collected. They were published while the patients still remained hospitalized. At the same time, in the COVID-19 vaccine FM, outcomes were reported in all cases. This allows to speculate that true mortality in COVID-19 FM may be higher.
4) Figure 1 does not look like a flow chart commonly used in the clinical study. We cannot follow the accurate sample number included or excluded by the criteria. The authors need to check clinical articles that have a flow chart of patient enrollment as a reference (for example, PMID: 30626332).
Response:
This is correct, the flow chart does not look like a flow chart in the clinical study - because we did not do a clinical study. We systematically reviewed the published literature on the subject, and our chart reflects the process of the paper selection, like in other systematic literature reviews done in accordance with the PRISMA standards - please see other similar papers e.g. PMID 30626332
5) Is there any special reason that the authors did not include the information on the medical treatment in Table 2? The authors need to keep consistency with Table 1.
Response:
The reason we included treatment into the Table 1 on COVID19 FM, but not is Table 2 (COVID vaccine-FM) is that there is antiviral treatment used for COVID infected patients but not for vaccine induced myocarditis. Nobody used any specific treatment for that other than vasopressors and inotropes or mechanical circulatory support, which we reported. To make the tables similar, we must remove the treatment column from Table 1.
6) What is the clinical implication of this study? The authors need to emphasize it.
Response:
We added the following paragraph:
It remains unclear, why most individuals, developing post-vaccine myocarditis, experience self-limited symptoms of chest pain and demonstrate mild troponin elevation, while other present with overwhelming hemodynamic compromise or cardiac arrest. The similarity of manifestations between COVID-19 FM and COVID-19 vaccine FM allows to suggest that such hyperreaction to either infection or the vaccine characterizes the individual reactivity more than the trigger. The devastating nature of fulminant myocarditis of either origin justifies future clinical, immunological, and genetic study in order to understand, treat, and ultimately prevent such response in the future.
Minor:
1) The conclusion of the abstract should be clear and concise. Detailed data and values should not be included in the section.
Response:
We rewrote the conclusions of the abstract to make it more concise and removed the numerical data. The new section reads as follows:
Conclusions: In the first series to retrospectively assess fulminant myocarditis associated with COVID-19 infection versus COVID-19 vaccination we found that both conditions had similarly high mortality rate, while COVID-19 FM has a more malignant course with more symptoms on presentation, more profound hemodynamic decompensation (higher heart rate, lower blood pressure), more cardiac arrests, and higher temporary MCS requirements including VA-ECMO. In terms of pathology, there was no difference with most biopsies/autopsies demonstrated lymphocytic infiltrates and some eosinophilic or mixed infiltrates. There was no predominance of young males in COVID-19 vaccine FM.
Moreover, we made multiple other revision in the abstract to make it more concise.
2) There is no information on the EMBASE in the abstract.
Response:
We overlooked this. Added.
3) I cannot understand the sentence of the abstract ‘The Student’s t-test was used for continuous variables and the χ2 statistic was used for categorical variables while the Wilcoxon Rank Sum Test was used for statistical comparisons’. T-test and χ2 statistics are also analysis for the comparison.
Response:
We re-phrased this to make the meaning clearer. It now reads as follows: The Student’s t-test was used for continuous variables and the χ2 statistic was used for categorical variables. For non-normal datadistributions, the Wilcoxon Rank Sum Test was used for statistical comparisons
4) There are some extra spaces between words and sentences.
Response:
We proof-read the manuscript and fixed the spaces
Reviewer 2 Report
Dear Authors,
i find this systematic literature review very interesting.
Even though COVID has less mediatic resonance nowadays, its myocardial interest is still a great topic. Even myocarditis related to COVID-vaccines is still a very interesting topic.
Manuscript is well written, full of citation from several authors and with a complete literature review of all most important articles about the topic.
Anyway, is your research from 1st December 2021(line 324) or 1st December 2020(line 106)?
Please discuss implications of results for practice and future research.
Author Response
Reviewer 2
Dear Authors,
i find this systematic literature review very interesting.
Even though COVID has less mediatic resonance nowadays, its myocardial interest is still a great topic. Even myocarditis related to COVID-vaccines is still a very interesting topic.
Manuscript is well written, full of citation from several authors and with a complete literature review of all most important articles about the topic.
Anyway, is your research from 1st December 2021(line 324) or 1st December 2020(line 106)?
Response:
We censored our search on August 31st 2022
Please discuss implications of results for practice and future research.
Response:
We added the following paragraph:
It remains unclear, why most individuals, developing post-vaccine myocarditis, experience self-limited symptoms of chest pain and demonstrate mild troponin elevation, while other present with overwhelming hemodynamic compromise or cardiac arrest. The similarity of manifestations between COVID-19 FM and COVID-19 vaccine FM allows to suggest that such hyperreaction to either infection or the vaccine characterizes the individual reactivity more than the trigger. The devastating nature of fulminant myocarditis of either origin justifies future clinical, immunological, and genetic study in order to understand, treat, and ultimately prevent such response in the future.
Round 2
Reviewer 1 Report
The authors revised almost all the problems I pointed out, however, there are still several points that need to be revised.
Major
1) The pathological type was identified in the limited cases of the COVID-19-induced myocarditis group (lower than 1/3 of patients). On the other hand, in almost all cases of the vaccine-induced group, the pathological type of myocarditis was identified. Authors need to mention this limitation in the discussion part of the manuscript.
2) I understand the author had decided to follow PRISMA. However, the authors should show the main group in the center and add the excluded group on the right side in each step (PMID: 29148960). The reader of the manuscript cannot understand how n=420 was reduced to n=151, and then to n=132. In addition, I cannot understand why ‘Record excluded (n=151)’ is located at the main flow, not on the right side. This is the point I want the authors to revise.
3) The author answered that ‘Nobody used any specific treatment for that other than vasopressors and inotropes or mechanical circulatory support, which we reported’. I checked articles the authors enrolled in the manuscript and found that some cases in the vaccine-induced myocarditis group received steroids as a treatment (ex PMID: 35088026). Adding this information will increase the value of the manuscript. In addition, I recommend the authors compare the frequency of using steroids in the two groups.
Minor
1) ‘There was no predominance of young males in COVID-19 vaccine’ in the last part of the abstract is new information for the reader of the manuscript that is not mentioned in the result part. In addition, this is not the primary outcome of the study. I recommend the authors remove it from the abstract.
2) I again recommend the authors to limit the contents of the discussion part of the abstract to make it clear and concise. For example, ‘more profound hemodynamic decompensation (higher heart rate, lower blood pressure), more cardiac arrests, and higher temporary MCS requirements including VA-ECMO’ can be summarized as ‘hemodynamic instability’. The authors have already described the details in the results section very well.
Author Response
The authors revised almost all the problems I pointed out, however, there are still several points that need to be revised.
Major
1) The pathological type was identified in the limited cases of the COVID-19-induced myocarditis group (lower than 1/3 of patients). On the other hand, in almost all cases of the vaccine-induced group, the pathological type of myocarditis was identified. Authors need to mention this limitation in the discussion part of the manuscript.
Response: Thanks for picking this up. We added the following statement to the Limitations section: Pathology was available only in some reports on COVID-19 FM, but almost in all cases of COVID-19 vaccine FM.
2) I understand the author had decided to follow PRISMA. However, the authors should show the main group in the center and add the excluded group on the right side in each step (PMID: 29148960). The reader of the manuscript cannot understand how n=420 was reduced to n=151, and then to n=132. In addition, I cannot understand why ‘Record excluded (n=151)’ is located at the main flow, not on the right side. This is the point I want the authors to revise.
Response: thank you for this comment, we revised and replaced the flow chart.
3) The author answered that ‘Nobody used any specific treatment for that other than vasopressors and inotropes or mechanical circulatory support, which we reported’. I checked articles the authors enrolled in the manuscript and found that some cases in the vaccine-induced myocarditis group received steroids as a treatment (ex PMID: 35088026). Adding this information will increase the value of the manuscript. In addition, I recommend the authors compare the frequency of using steroids in the two groups.
Response: This s correct. We added the column to Table 2 where we indicated if systemic steroids were used (in act, we replaced the Table 2).
Minor
1) ‘There was no predominance of young males in COVID-19 vaccine’ in the last part of the abstract is new information for the reader of the manuscript that is not mentioned in the result part. In addition, this is not the primary outcome of the study. I recommend the authors remove it from the abstract.
Response: We gave this a careful consideration. In fact, this is a very unexpected and unusual finding. To underscore this, we added the following sources:
In published reports on COVID-19 vaccine induced myocarditis, males constituted over 65% (90); male to female ratio was reported as 3:2 (91) or even 14:1 (92). By another report, male sex increased the odds of myocarditis 4.4 times (93). In our review, only 40.1% of patients were male, which also suggests a principally different mechanism of FM comparing to usual clinically mild myocarditis.
Taking these reports into consideration, our preference would be to leave this finding.
2) I again recommend the authors to limit the contents of the discussion part of the abstract to make it clear and concise. For example, ‘more profound hemodynamic decompensation (higher heart rate, lower blood pressure), more cardiac arrests, and higher temporary MCS requirements including VA-ECMO’ can be summarized as ‘hemodynamic instability’. The authors have already described the details in the results section very well.
Response: Thank you or this comment. We shortened the discussion.